# Evaluating Cement Treated Aggregate Base Containing Steel Slag: Mechanical Properties, Volume Stability and Environmental Impacts

**DOI:** 10.3390/ma15238277

**Published:** 2022-11-22

**Authors:** You Huang, Xin Yang, Shuai Wang, Zhaohui Liu, Li Liu, Bo Xu

**Affiliations:** 1Engineering Research Center of Catastrophic Prophylaxis and Treatment of Road & Traffic Safety of Ministry of Education, Changsha University of Science & Technology, Changsha 410114, China; 2School of Traffic and Transportation Engineering, Changsha University of Science and Technology, Changsha 410114, China; 3Power China Guiyang Engineering Co., Ltd., Guiyang 550000, China

**Keywords:** steel slag, cement-stabilized aggregate, base course, mechanical properties, volume expansion, environmental impact

## Abstract

Steel slag has been commonly used in road engineering as cementitious material; however, its application in base course is not widely reported. Four contents of steel slag (0%, 30%, 50%, 75% by volume) were blended into different cement (3%, 4%, 5%, 6% by weight)-treated aggregates. Mechanical properties, volume stability, economic benefits and environmental influences of steel slag mixtures were investigated for the feasibility of applying steel slag in semi-rigid base course. Abrasion, crushing and elongated particle content were compared against limestone aggregate, showing that steel slag has the potential of replacing natural aggregate in concrete. Steel slag is beneficial for reinforcement of the strength and stiffness. The mixture has the highest strength and stiffness when bended with 50% steel slag at 4% cement content. By treating steel slag with CH_3_COOH or adding silica fume, volume expansion of steel slag can effectively be controlled. Larger size steel slag (>4.75 mm) and higher cement content are recommended due to heavy metal leaching risk, especially in salty humid areas. Steel slag has sound economic benefits due to the relatively low price. Environmental benefits can also be achieved given that the transport CO_2eq_ emission of steel slag is accounted for. With proper control in production process, steel slag is a very promising alternative material to be utilized in cement-stabilized base course in road engineering.

## 1. Introduction

Road infrastructure in China has experienced a fast development in the past three decades. The total mileage of motorway is over 5 million kilometers at the end of 2021, including nearly 160 thousand kilometers of high-speed expressway [1]. In the meantime, about 800 thousand kilometers of motorway needs rehabilitation or maintenance each year. A large number of aggregates are required to meet the needs of new constructions and maintenances, rendering a huge burden on natural resource and environment protection. There are urgent needs for sustainable and environmental substitute for natural aggregates in road engineering. At the same time, industrial wastes have increased as a result of rising population and technological development, posing increasingly serious social problems and environmental threats. Innovative approaches have been studied to decrease these wastes, or, as a clearer option, turn them into valuable reserves [2,3]. For several decades, various industry wastes have been extensively studied to replace aggregates or to enhance performance of construction materials [4,5,6].

Steel slag is one of the by-products generated in steel making process. China’s crude steel production reaches 1.03 billion tons in 2021 [1]. According to production rate of 60~150 kgt^−1^ based on the current refining processes, the steel slag production is around 60~150 million tons yearly. A huge quantity of steel slag needs to be handled properly. Nevertheless, compared to 85~98% of utilization rate in industrialized countries such as USA, Europe and Japan, the utilization in China is only at 29.5% [7]. Most of it is piled up or disposed as landfill, which not only occupies land resource, but also causes leaking of harmful component. 

The physic properties of steel slag, i.e., hardness, crushing and grinding, makes it adequate for road engineering. If used appropriately, steel slag can partially or fully replace aggregates in mixtures; not only can the shortage of construction resources be alleviated, but also the damage to environment can be reduced. The utilization of steel slag for road construction in the United States is at nearly 50%; in Europe at 43%, and in Japan at 32.4%, whilst in China it is only at 7.6% [7]. Much work has been done to apply steel slag in asphalt pavement in recent years. He et al. and Skaf et al. have argued that road construction is the most effective and lowest-cost way to utilize steel slag [8,9]. Studies have suggested that electric steel slag displayed almost the same mechanical and physical properties as traditional aggregates [10,11]. Scholars have also found that steel slag can improve high temperature performance [12,13], moisture stability [14] and fatigue [15] of asphalt mixture significantly. Moreover, the solutions to obtain functional pavement such as improving skid resistance [16], deicing [17] and self-healing material [18] became more vital and diverse with the application of steel slag. In addition, the utilization of steel slag mixtures has sound economic benefit and potential environmental benefits compared to traditional mineral aggregate mixtures [19,20,21]. Despite the many merits of steel slag over natural aggregates, the applications of steel slag in paving material are still conservative. Concerns include physical and morphologic property, replacement content, long-term volume stability and environmental impacts [22,23,24]. 

Previous studies on steel slag in road engineering mainly focus on asphalt mixture and surface course, relatively few on applications of incorporating steel slag in base layer. Based on the reality that cement-stabilized base is the main base type, and the massive slag waste in China at this stage and beyond, the utilizing of steel slag in cement-stabilized base was investigated in this study. We designed four levels of replacement: 0%, 30%, 50% and 75% of steel slag, and the optimal moisture contents were determined through compaction test. Performances of samples with different steel slag contents were evaluated. Unconfined compressive strength, bending strength and resilient modulus were tested for mechanical properties. Volume expansion of steel slag and dry shrinkage of steel slag mixture were also checked. Environmental impacts were evaluated in view of heavy metal leaching and carbon footprint. Other means including SEM, X-ray and FAAS were also employed to check the surface morphology, element distribution and chemical composition to comprehend the strength development and environmental impacts of steel slag in cement-stabilized semi-rigid base. 

## 2. Materials and Specimen

### 2.1. Materials

The steel slag used in this study was obtained from Shenglong Steel Mill and the limestone was from a quarry, in Fangchenggang, Guangxi, China, as displayed in Figure 1. Cement of 42.5 MPa was produced by Runfeng. Physical and mechanical properties of steel slag and limestone are compared in Table 1. As properties of steel slag would change with aging time, samples were examined at different aging times, namely, 0 months, 6 months and 12 months. Although decreasing slightly with aging time and varying with particle size, the apparent density of steel slag is about 25~36% higher than that of limestone. At the same time, crushing, abrasion and elongated particle content of steel slag are smaller than those of limestone. Water absorption of steel slag is higher, which can be explained by the porous surface shown in the multi-scale images by Scanning Electron Microscopy (SEM) in Figure 2. However, the water absorption decreases with aging time. Table 2 records the chemical compositions of steel slag through an X-ray fluorescence semi-quantitative analysis. The most prominent component is CaO, followed by Fe_2_O_3_ and SiO_2_, accounting for about 84% altogether. It is notable that f-CaO and f-MgO experience steady declines with aging time. Metal constitutes of the steel slag were detected with grinded steel slag powder (<0.125 μm) by the Flame Atomic Absorption Spectrophotometer (FAAS). Heavy metal elements of steel slag with 12 months of aging are displayed in Figure 3. Mn, Cr and V are the top 3 heavy metals. 

### 2.2. Mixture Design and Specimen Preparation

The particle sizes of steel slag on site mainly lie between 4.75 mm~19 mm, as Figure 1 shows. Due to the differences of density between steel slag and limestone, volume was chosen as an indicator for steel slag replacement. Thus, a volume–mass conversion was performed on aggregates of steel slag and limestone. The composite gradation of “steel slag + limestone” mixture in 0.45 power chart is illustrated in Figure 4. 

Cement-treated aggregates with four steel slag contents of 0% (control group), 30%, 50%, 75% and four cement contents of 3%, 4%, 5%, 6% were formulated. The samples of different steel slag contents were compacted using impaction test method as per JTG E51-2009 [25] to determine the relationship between water content and dry density. The determined optimum moisture content (OMC) and maximum dry density (MDD) are listed in Table 3. With steel slag content growing, the OMCs and MDDs of mixture increase. OMC of samples with 50% steel slag is about 30% higher than that of the control group. 

The cylindrical specimens with dimension of Φ 150 mm × 150 mm were adopted for unconfined compressive strength (UCS), unconfined compressive resilient modulus (UCRM) and volume expansion tests. Prism specimens with dimension of 100 mm × 100 mm × 400 mm were used for bending strength (BS) and dry shrinkage tests. The samples were controlled at 98% compactness and at least four duplicates were prepared for each test. Samples for UCS and dry shrinkage test were cured in standard condition (temperature of 20 ± 1 °C, humidity of 95 ± 5%) for 7 days, while samples for UCRM and BS test were cured for 28 days. The production and curing process of samples are shown in Figure 5.

## 3. Methodology

### 3.1. Mechanical Test

To investigate the influence of steel slag on strength and stiffness, the UTM-100 servo-hydraulic multi-functional material test system was employed to test unconfined compressive strength (UCS), bending strength (BS) and unconfined compressive resilient modulus (UCRM). When the samples were subjected to UCS test, the loading rate was set to 1 mm/min, while in the BS test, the loading rate was set to 50 mm/min, as per JTG E51-2009 [25]. For the UCRM test, the specimens were subjected to five loading levels, with the maximum load of 65% of the strength, as per JTG E51-2009 [25]. The mechanical tests are shown in Figure 6.

### 3.2. Volume Stability Test

The volume stability test consists of two parts: the volume expansion of steel slag and the dry shrinkage of cement-stabilized aggregates containing steel slag. The safety of highway is affected by the possible expansion caused by f-CaO and f-MgO in steel slag. Steel slag with no aging was compacted at OMC, and soaked in water in the mold with permeable platters at each end. Dial directors were set to monitor the volume increase for 10 consecutive days. Samples treated with CH_3_COOH or silica fume were also tested to clarify their inhibition effect on volume expansion. To explore the shrinkage of mixture with different steel slag contents, after curing in standard condition for 7 days, prism samples were examined for initial mass and length and then set in a condition of 20 ± 1 °C and 60 ± 5 % humidity with both ends measured with dial directors to monitor the length change due to dry shrinkage. The mass and length of samples were recorded every day during the first 7 days, every other day from 7th to 28th day, and on the 60th and 90th day. In the end, the specimens were heated to constant weight. Length and mass of the final state were measured. The volume expansion of steel slag and dry shrinkage of cement concrete with steel slag are in Figure 7.

### 3.3. Heavy Metal Leaching Test

The heavy metal pollution of steel slag was examined through precipitation of Cr and V. Influence of particle size, cement content and water salinity were evaluated. An amount of 100 g of steel slag with different particle size from less than 0.075 mm to larger than 26.5 mm (Figure 8a) was put into 200 mL pure water for 24 h. To check the influence of cement content on metal precipitation, samples of Φ 150 mm ×150 mm containing 50% steel slag with 4%, 5% and 6% cement content (Figure 8b) were prepared and put into 3250 mL pure water to be fully emerged. A control group with no steel slag was examined as well. As Fangchenggang is a coastal city in southern China, heavy metal precipitation in seawater environment was also checked. To eliminate size effect, the steel slag aggregates were grinded into powder, and 100 g samples were put in 200 mL water of 0%, 0.5%, 1%, 1.5%, 2%, 2.5% salinity (Figure 8c). Change of heavy metal in the water was detected every 4 h for a total duration of 24 h.

### 3.4. Economic Evaluation and Carbon Footprint Simulation

A case study of cement-stabilized aggregate base course for a typical four-lane highway with a dimension of 1 km long, 20 cm thick and 32 m wide was conducted to investigate economic gains and greenhouse gas (GHG) emissions. In the definition of cement-treated aggregates, mainly Portland cement and gravel constitute the composition. Water is not considered in the subsequent calculation due to its small amount. The consumptions of steel slag, limestone aggregate and cement were determined based on the compaction experiments (Table 3). Market prices of raw materials were surveyed. In the transportation cost calculation, compared to the commonly seen natural aggregate quarries located in many parts of the country, steel slag appears only in fixed steel mills. Silica fume is also hauled long distance from manufactories. As a result, the average transport distance of 40 km for cement, 40 km for limestone, 100 km for steel slag and silica fume were assumed.

To further compare the environmental impacts of cement-stabilized aggregate with steel slag in the aspect of equivalent CO_2_ (CO_2eq_) emission, life cycle analysis (LCA) models of cement-stabilized aggregate base course were prepared with SimaPro 9 software. The methodology of LCA consists of 4 stages including goal and scope of the project, inventory analysis, impact assessment and interpretation of results. Correspondently, the research methodology followed for this research is represented in Figure 9. Consumption of natural resources and CO_2eq_ emission for mix preparation is considered the project’s goal and scope. In inventory analysis, binder materials and aggregates are considered from the Ecoinvent database as per European standards. In the LCA assembly stage, for mix preparation, all the binders, minerals and steel slags are evaluated as per the mix compaction as indicated in Table 3. Since steel slag is recycled as a by-product of steelmaking, the energy consumption of producing steel slag is zero. The only major environmental impact of steel slag is the transport process. Table 4 collects the energy consumption data for cement and limestone production life cycle analysis (LCA) [19]. In the impact assessment, the ReCiPe 2016 midpoint method was used in SimaPro for LCA [26,27,28]. The mid-point implications include climate change challenges, human toxicity, loss of the ozone layer, acidification, and abiotic capital depletion, and is classified as more precise. Electricity was the final contribution to the mix phase. Energy is necessary to blending the ingredients to obtain the end product. The shipping of the material was also taken into account, by 32-ton trucks. The average transport distance of 40 km for cement, 40 km for limestone, 100 km for steel slag and silica fume were assumed. Global and European database values available in SimaPro 8.4 were used to characterize and normalize the environmental impacts of the products.

## 4. Results and Discussions

### 4.1. Mechanical Properties

The UCS (Figure 10a), BS (Figure 10b) and UCRM (Figure 10c) are overall positively related to cement content and steel slag: with steel slag replacement content increasing, UCS and BS keep climbing until reaching its peak at 50% replacement, followed by a small decrease. The average UCS of 4% cement treated aggregates without steel slag is 3.73 MPa, comparatively, UCS of specimens with 30% steel slag is 4.33 MPa, about 15% higher, met the strength requirement of cement-stabilized base for heavy traffic pavement (4.0~6.0 MPa) [29]. Especially when steel slag content is at 50%, the UCS is around 5.4 MPa, meeting the strength requirement of cement-stabilized base for extremely heavy traffic pavement (5.0~7.0 MPa) [29]. In other words, steel slag could be applicated in cement-stabilized base for heavy traffic pavement.

In general, when steel slag content increases from 0 to 50%, the strength and stiffness increases. This is because steel slag is a very strong material with rough surface that strengthens the particle extrusion, which helps form a strong aggregate skeleton to increase the resistance to damage. Moreover, the cementitious components of C_3_S/C_2_S contained in steel slag are hydrated to obtain C-S-H gel [30], which constitutes the fundamental building block of hydrated cement and makes the system stronger:2(3CaO·SiO_2_) + 6H_2_O→3CaO·2SiO_2_·3H_2_O + 3Ca(OH)_2_,(1)
2(2CaO·SiO_2_) + 4H_2_O→3CaO·2SiO_2_·3H_2_O + Ca(OH)_2_.(2)

In the event that gypsum is present in cement clinker, C_3_A would also react with CaSO_4_ to form trisulfide calcium aluminate hydrate (C_6_AS_3_H_32_). The internal voids of cement slurry and surface porosity of steel slag are filled with these two hydration products, and the bonding effect of cement particle is improved. Apparently, the cementitious components contained in steel slag would help increase the hydration products, thereby reinforcing the mechanical performance and durability of the mixture.

However, when the content of steel slag further increases, the high porosity of steel slag increases the voids in the sample. Some of the cement would be adsorbed and wrapped in the surface porosity of steel slag and insulate from further hydration reaction, which is unbeneficial to the strength formation. In this study, when steel slag increases from 50% to 75%, the 7d UCS experienced a decline of 6~14%, and 28 UCRM a decline of 14~18%, indicating that the adverse effects of porosity began to overtake the positive effects of rough surface and gelling composition when replacement content of steel slag is too high. The recommended steel slag replacement is about 50% based on the test results in this study.

### 4.2. Volume Stability

#### 4.2.1. Volume Expansion of Steel Slag

The volume changes of steel slag are displayed in Figure 11. As is shown, the volume of steel slag increases sharply with immerging time. This is mainly due to the reaction of f-CaO contained in steel slag with water. The volume expansion can be significantly reduced by treating the steel slag with CH_3_COOH or adding silica fume. As can be seen, the volume expansion has a negative relation with CH_3_COOH concentration or silica fume content. In this study, volume expansion of steel slag treated with 15% CH_3_COOH or 4.8% silica fume is reduced very close to 0%. However, the expansion rises again when silica fume is increased to 9.6%; the excessive silica fume downgrades the skeleton structure of aggregate and decreases the interlock of aggregates. To assure the volume stability in the long run, it is recommended that the content of silica fume should be strictly controlled, preferably around 4.5%.

To ensure long-term performance, there are several recommendations to control the possible expansion of steel slag: Firstly, steel slag should be detected for active substances (free calcium oxide or free magnesium oxide). Unreliable source of steel slag with high content of f-CaO or f-MgO should not be used. Secondly, steel slag needs to be treated to mitigate the expansion effects, either by aging for a period of time (as indicated that f-CaO or f-MgO of steel slag steadily decline with aging time in Table 2), or treated with appropriate CH_3_COOH or silica fume. Moreover, adding silica fume to steel slag during production process when it is still in liquid molten state is an effective solution [21].

#### 4.2.2. Shrinkage of Steel Slag Mixture

The recorded weight and length change of cement-stabilized mixes were employed to calculate the moisture loss and shrinkage coefficients (Equations (3) and (4)).
(3)$$\omega_{i}=\frac{m_{i}-m_{i+1}}{m_{c}}$$
(4)$$\alpha_{d}=\frac{\sum \varepsilon_{i}}{\sum \omega_{i}}$$
where *ε_i_* is the shrinkage strain; *ω_i_* is the moisture loss rate; *m_i_* is the specimen weight; *m_c_* is the constant dry weight of specimen; *α_d_* is the shrinkage coefficient.

Cumulative moisture loss rates and shrinkage coefficients and represented in Figure 12. The moisture loss increases fast for the first few days because of free water evaporation. The moisture loss at stable stage (after 30 days) of samples with 50% steel slag is much higher than those with no steel slag, reflecting the fact that higher steel slag content corresponds to higher water loss due to the higher water content. This reminds us that more attention should be given to the moisture maintenance of cement-stabilized base with steel slag in the first 1~2 weeks to reduce moisture loss. Then, it began to climb slowly. This stage is the gradual bound water loss with time. After about 30 days, the water loss becomes stable. It is found that steel slag inhibits the drying shrinkage: the shrinkage coefficients decrease at higher steel slag content. The final shrinkage coefficients of samples with 50% of steel slag are about 40% less than the control group. It is also evident in previous research that the expansion caused by f-CaO and f-MgO contained in steel slag may counteract partial shrinkage [31].

### 4.3. Heavy Metal Leaching

Cr and V leaching of different particle sizes are listed in Table 5 and Table 6. It is clearly shown that heavy metal leaching of small particle size is much higher than that of large ones. Cr precipitations of 4.75~9.5 mm, 9.5~19 mm, 19~26.5 mm particles are 60%, 67% and 71% lower than of those passing 0.075 mm, respectively. V precipitations of 4.75~9.5 mm, 9.5~19 mm, 19~26.5 mm particles are about 40%, 60% and 64% lower than of those passing 0.075 mm, respectively. Small particle size has higher specific surface area, meaning more contact area with water. In this sense, it is recommended that steel slag used in cement-stabilized base be not smaller than 4.75 mm.

Precipitations of Cr and V with different cement content are shown in Figure 13. The precipitation of heavy metal decreased significantly after being mixed with cement, as more cement mortar provides better coverage and inhibits heavy metal from leaching into surrounding environment. Compared to 4% cement, precipitation of Cr and V at 6% cement declined about 25% and 35%, respectively.

The precipitation of heavy metal was also checked in salty water. The salinity of sea water nearby Fangchenggang is about 2%. Figure 14 demonstrates that 24 h cumulative precipitation of Cr and V at 2% salt water are nearly threefold and double those in pure water, respectively. The active ions contained in salty water can react with the metal ions in steel slag, and also would corrode the covering cementitious material which in turn promotes heavy metal precipitation. When utilized in salty water environment, stricter measurements should be implemented to prevent possible heavy metal pollution.

### 4.4. Economic and Carbon Footprint Analysis

Steel slag has shown sound performance following similar procedures of pavement engineering; the economic cost and greenhouse gas (GHG) emission are two other aspects to check. The material and transport costs were calculated for 50% and 30% steel slag replacement, as shown in Table 7. Due to the low price of steel slag, the total costs of steel slag mixture can be saved approximately by about 22% (50% steel slag replacement) and 15% (30% steel slag replacement). Even with 3% silica fume added, the total costs can still be economized by 12% and 10%, respectively, indicating that the steel slag has sound economic gains.

The CO_2eq_ emission of different steel slag contents were displayed in Figure 15, Figure 16, Figure 17 and Figure 18. As shown in Figure 15, Figure 16 and Figure 17, for 1 P of cement-stabilized limestone aggregate base course (1 km long by 20 cm thick by 32 m wide), the CO_2eq_ emission is 3.06 × 10^5^ kg. Comparatively, the CO_2eq_ emissions of 1 P of “30% steel slag +70% limestone” and “50% steel slag + 50% limestone” base course are 3.18 × 10^5^ kg and 3.37 × 10^5^ kg, respectively. It is found that in the initial assumption of 100 km transport distance for steel slag the total CO_2eq_ emissions of “steel slag + limestone” mixtures actually are higher than the traditional limestone concrete. The corresponding CO_2eq_ emissions for each procedure in the mixture production are summarized in Figure 18. Cement production contributes the most CO_2eq_ emission, accounting for nearly 60% of the total emission. This adheres to our ordinary expectation that energy consumption and pollution from cement production is very high. Unlike the overall life cycle, economic performance is usually less sensitive to transport distance [32], CO_2eq_ emission from material transport is non-neglectable, and it grows with steel slag replacement. For instance, the CO_2eq_ emission from material (mainly steel slag) transport accounts for as high as 30% of the total emission for the “50% steel slag + 50% limestone” mixture. The environmental impacts of steel slag depend on the weight of steel slag as well as its haul distance. In this sense, we further calculated the critical transport distance. The critical transport distance is 59 km for “50% steel slag + 50% limestone” mixture, and 74 km for “30% steel slag +70% limestone” mixture. That is, environmental benefits can be achieved if the transport distance is below 59 km for “50% steel slag + 50% limestone” mixture, and below 74 km for “30% steel slag + 70% limestone” mixture.

## 5. Conclusions

Steel slag is a promising alternative for natural aggregates in pavement base course. In this study, different steel slag content (0%, 30%, 50%, 75%) was added to cement-treated aggregates. Throughout investigations including mechanical properties, volume characterization, economic benefits and environmental impacts, main conclusions are as follows:(1)Compared to traditional mineral aggregate (i.e., limestone), steel slag aggregate has equal or superior engineering properties, including density, abrasion resistance and crushing resistance. Due to higher water absorption and larger gravity, the OMC and MDD increase proportionally as more natural aggregates are replaced with steel slag.(2)Mechanical properties including strength and stiffness are significantly improved by steel slag within 50% replacement due to the increased hydration products and interlocked aggregate structure. However, the porosity and water absorption rate have greater negative effect on strength when steel slag replacement further increases.(3)Appropriate treatment with CH_3_COOH or addition of silica fume can relieve the volume expansion of steel slag. Nevertheless, excessive silica fume may be detrimental to the aggregate structure and should be kept within a reasonable content: 3–4.5%.(4)Smaller particle size shows higher risk of heavy metal precipitation, and the salty water environment in coastal area adds to it. Large particle size (>4.75 mm) and higher cement content are recommended to reduce heavy metal leaching threat.(5)Steel slag mixture has sound economic gains and potential environmental benefits compared to limestone mixture. However, the transport emission of steel slag should be accounted for to determine an environmental haul distance of steel slag.

Further research would be focused on long-term performances of cement-treated aggregate base containing steel slag in real pavement projects in both the theoretic and engineering aspects.

## Figures and Tables

**Figure 1 materials-15-08277-f001:**
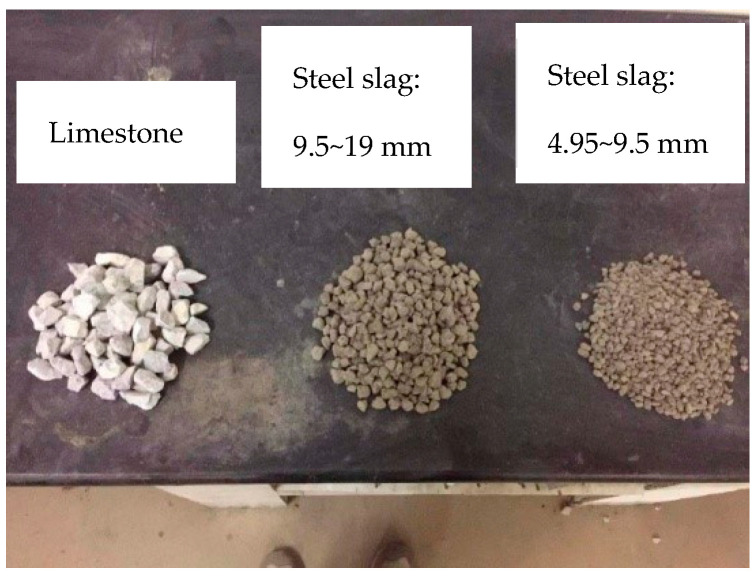
Aggregates of limestone and steel slag.

**Figure 2 materials-15-08277-f002:**
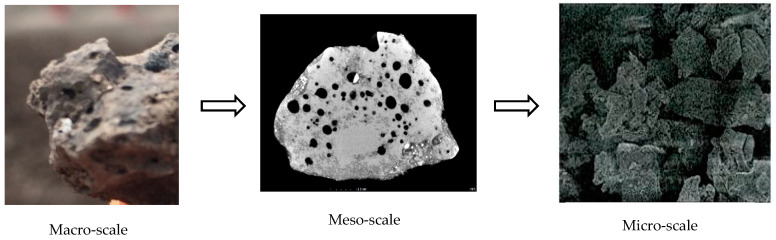
Multi-scale structure of steel slag by SEM.

**Figure 3 materials-15-08277-f003:**
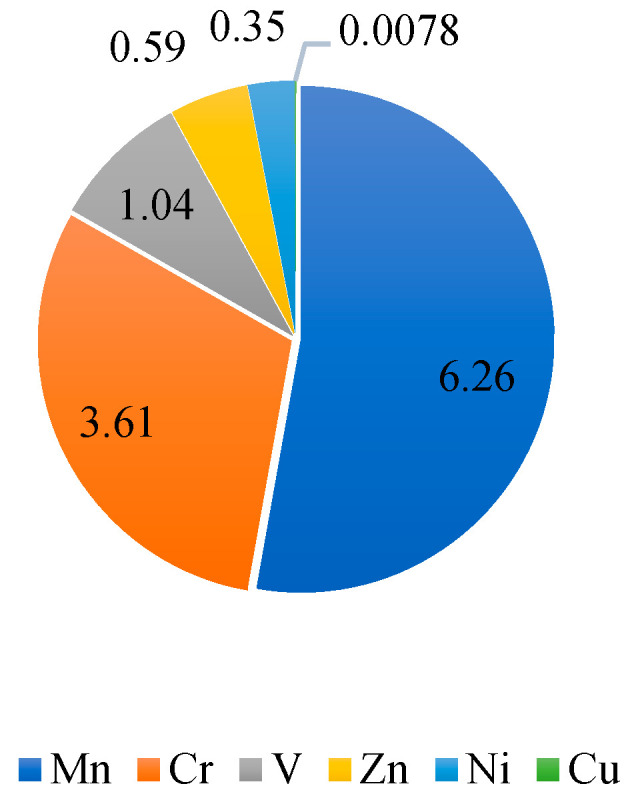
Heavy metal components of steel slag.

**Figure 4 materials-15-08277-f004:**
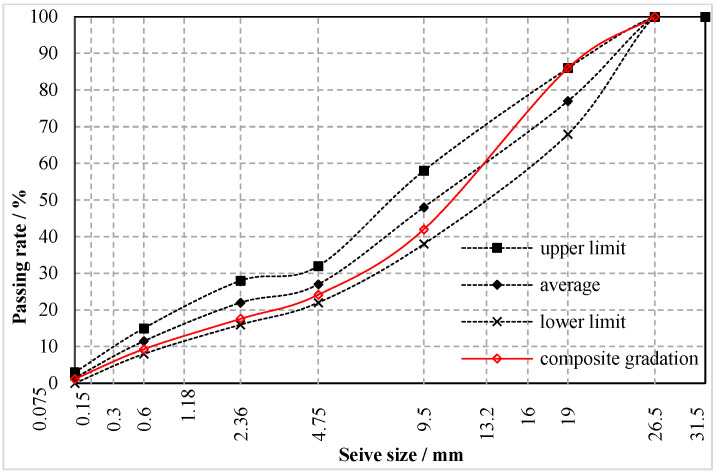
Mixture gradation.

**Figure 5 materials-15-08277-f005:**
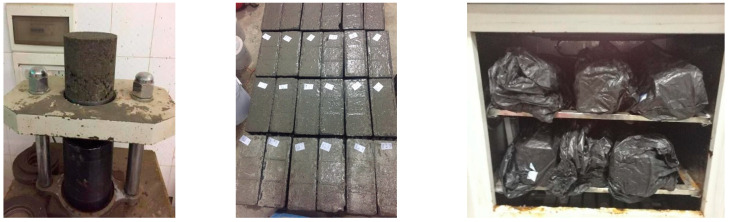
Specimen preparation and curing.

**Figure 6 materials-15-08277-f006:**
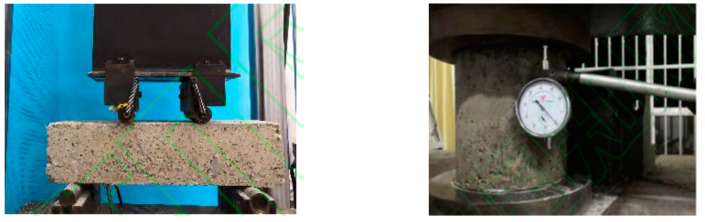
Mechanical properties test.

**Figure 7 materials-15-08277-f007:**
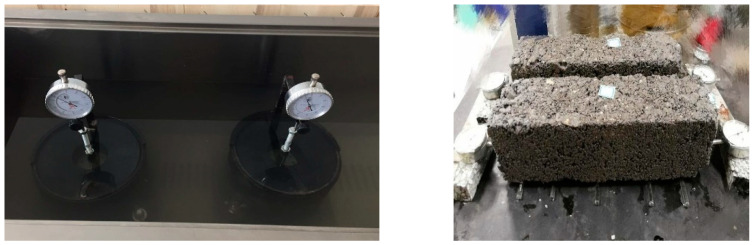
Volumetric properties test.

**Figure 8 materials-15-08277-f008:**
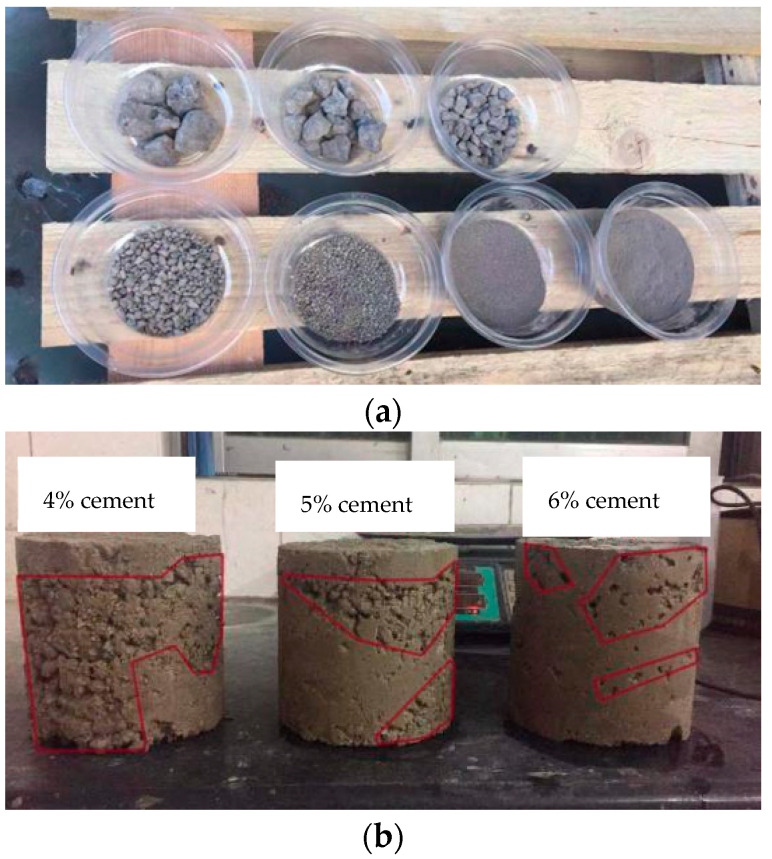
Heavy metal leaching test. (**a**) Steel slag aggregates of different particle size; (**b**) Samples with different cement content; (**c**) Containers with different seawater salinity.

**Figure 9 materials-15-08277-f009:**
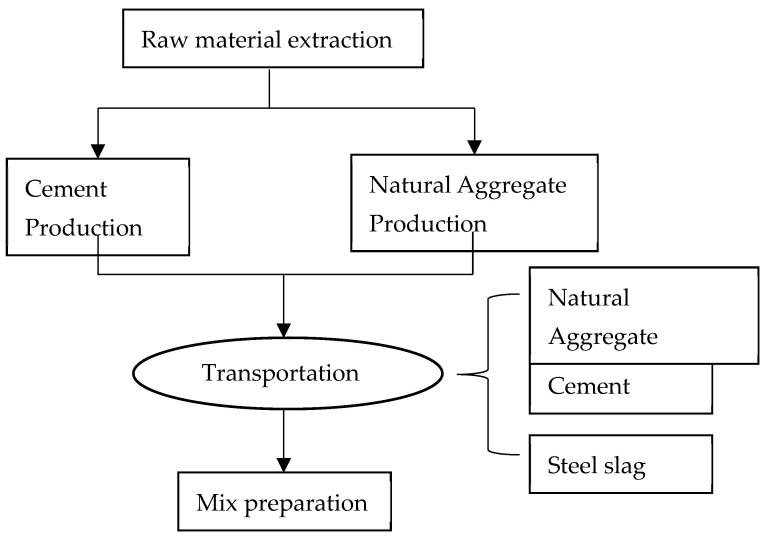
Virtual process for the cement-stabilized aggregate mix preparation.

**Figure 10 materials-15-08277-f010:**
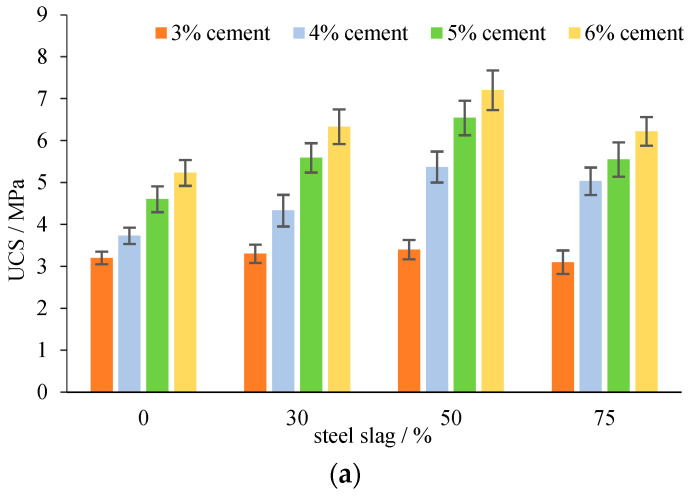
Mechanical properties of mixtures with steel slag. (**a**) UCS at 7 d; (**b**) BS at 28 d; (**c**) UCRM at 28 d.

**Figure 11 materials-15-08277-f011:**
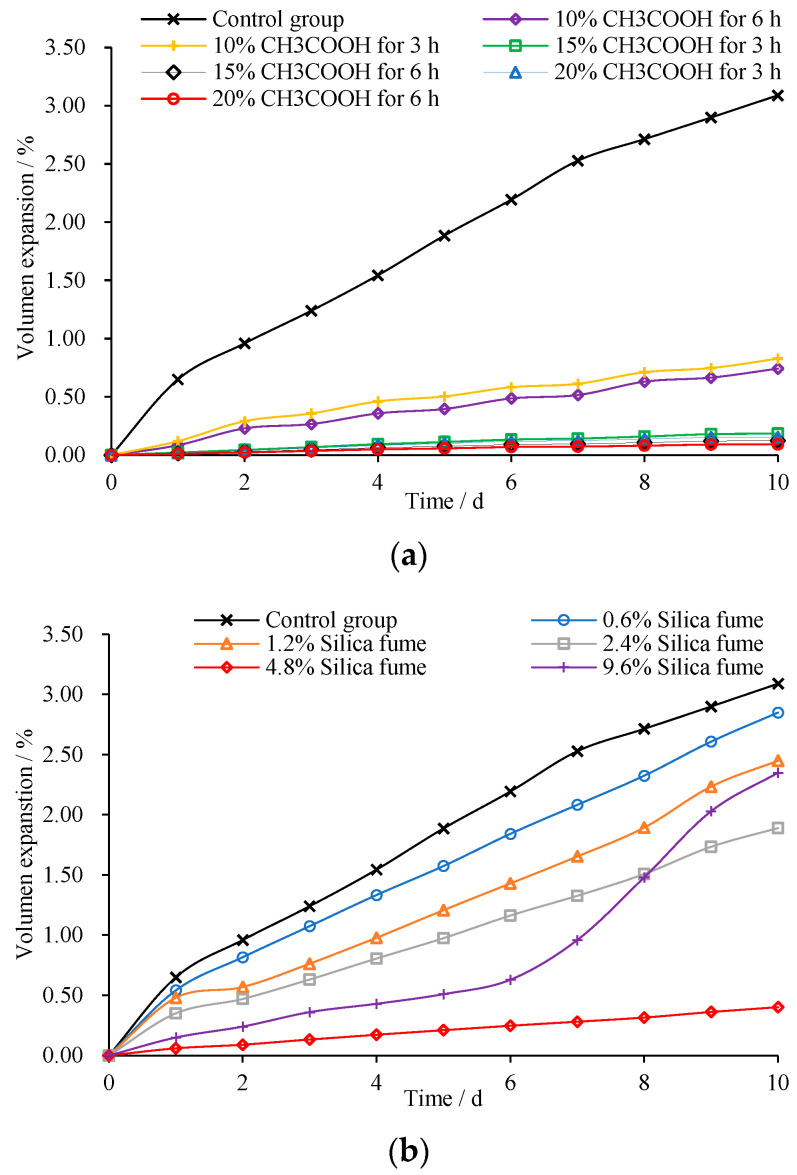
Volume stability results. (**a**) Treated with CH3COOH; (**b**) Addition of silica fume.

**Figure 12 materials-15-08277-f012:**
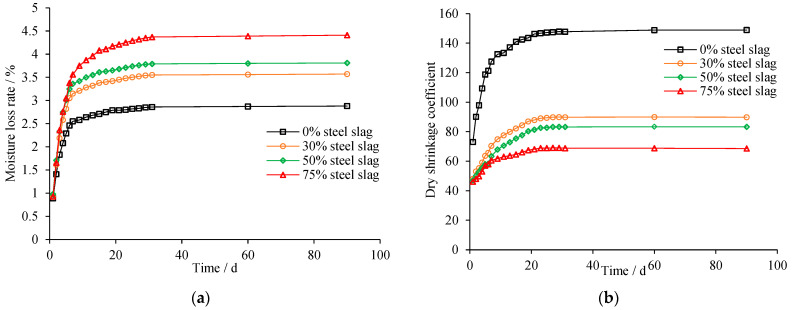
Shrinkage of mixtures with different steel slag content (4% cement content). (**a**) Moisture loss; (**b**) Shrinkage coefficient.

**Figure 13 materials-15-08277-f013:**
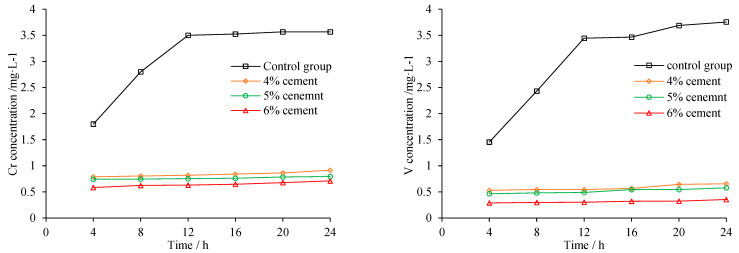
Heavy metal precipitation of different cement content.

**Figure 14 materials-15-08277-f014:**
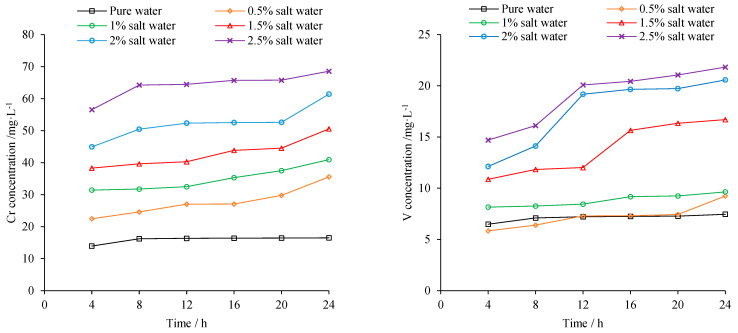
Heavy metal precipitation at different salinity.

**Figure 15 materials-15-08277-f015:**
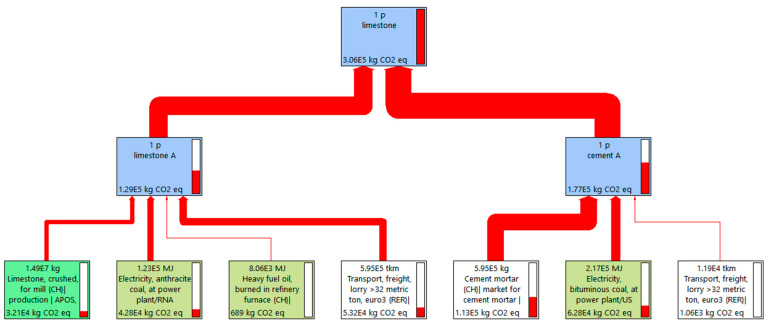
CO_2eq_ emission of 100% limestone mixture.

**Figure 16 materials-15-08277-f016:**
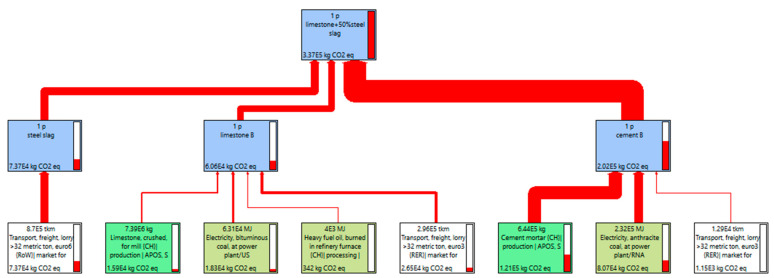
CO_2eq_ emission of 50% steel slag + 50% limestone with transport distance 100 km.

**Figure 17 materials-15-08277-f017:**
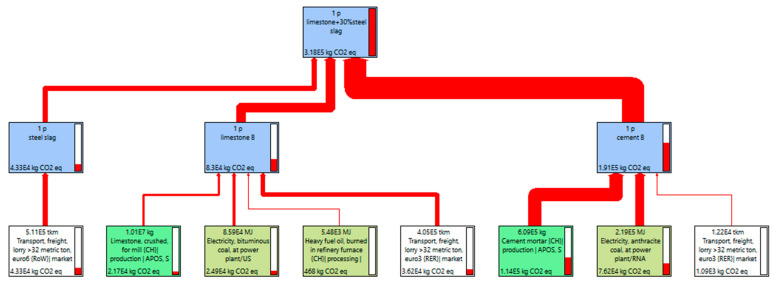
CO_2eq_ emission of 30% steel slag + 70% limestone with transport distance 100 km.

**Figure 18 materials-15-08277-f018:**
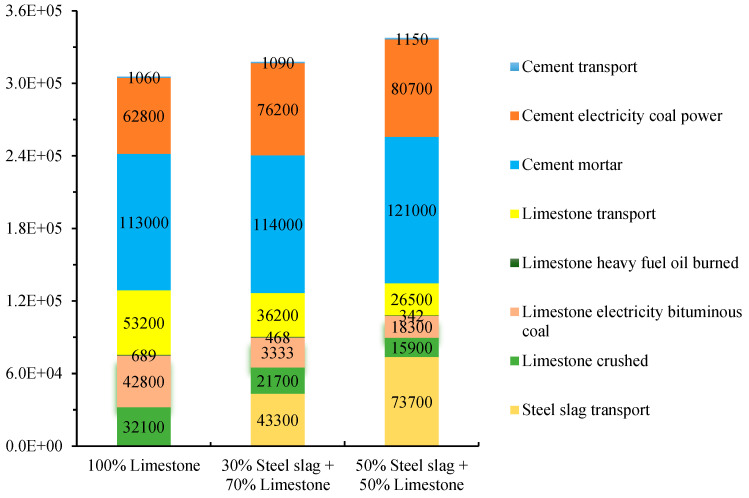
Detailed CO_2eq_ emissions for each procedure.

**Table 1 materials-15-08277-t001:** Physical and mechanical properties of steel slag and limestone.

Index		Steel Slag			Limestone
		Aging 0 Month	Aging 6 Months	Aging 12 Months	
Apparent density (g/cm^3^)	4.75~9.5 mm	3.551	3.482	3.374	2.661
	9.5~19 mm	3.556	3.496	3.387	2.641
	19~26.5 mm	3.555	3.475	3.397	2.667
Crushing value (%)		17.4	18.4	19.2	25.4
Abrasion (%)		12	12.5	13	22.7
Elongated or flaky particle content (%)		4.6	3.9	3.2	5.3
Water absorption (%)	0.5 h	1.10	1.08	1.01	0.29
	1 h	2.25	2.22	2.11	0.61
	2 h	2.45	2.41	2.30	0.64
	4 h	2.54	2.48	2.33	0.71
	8 h	2.66	2.60	2.41	0.72
	12 h	2.76	2.68	2.44	0.80
	24 h	2.96	2.87	2.53	0.92
	48 h	3.08	2.95	2.69	1.02

**Table 2 materials-15-08277-t002:** Chemical composition of steel slag (%).

Oxide Type	CaO	Fe_2_O_3_	SiO_2_	MgO	Mn	Al_2_O_3_	P_2_O_5_	f-CaO	f-MgO	Others
Aging 0 month	48.26	19.01	16.41	6.6	3.74	3.73	1.23	3.42	0.037	1.02
Aging 6 month	44.5	20.62	17.86	6.43	3.56	3.54	1.65	2.43	0.034	1.84
Aging 12 month	40.74	22.23	19.31	6.26	3.38	3.35	2.07	1.44	0.028	2.66

**Table 3 materials-15-08277-t003:** OMC and MDD of different steel slag content and cement content.

Combinations	OMC (%)	MDD (g/cm^3^)
0% steel slag + 3% cement	3.98	2.237
0% steel slag + 4% cement	4.10	2.325
0% steel slag + 5% cement	4.32	2.414
0% steel slag + 6% cement	4.87	2.498
30% steel slag + 3% cement	4.11	2.344
30% steel slag + 4% cement	4.31	2.409
30% steel slag + 5% cement	4.55	2.450
30% steel slag + 6% cement	5.17	2.542
50% steel slag + 3% cement	5.01	2.503
50% steel slag + 4% cement	5.37	2.544
50% steel slag + 5% cement	5.62	2.579
50% steel slag + 6% cement	6.27	2.599
75% steel slag + 3% cement	6.97	2.606
75% steel slag + 4% cement	7.21	2.620
75% steel slag + 5% cement	7.38	2.641
75% steel slag + 6% cement	8.11	2.653

**Table 4 materials-15-08277-t004:** Energy consumptions for unit material production [19].

Materials	Electricity (MJ/kg)	Diesel (MJ/kg)
Cement	0.36	-
Limestone	0.00828	0.000542
Steel slag	-	-

**Table 5 materials-15-08277-t005:** Cr precipitation (mg/L).

Time	<0.075	0.075~0.6	0.6~2.36	2.36~4.75	4.75~9.5	9.5~19	19~26.5
4 h	7.945	6.214	4.997	4.103	3.412	2.845	2.217
8 h	8.213	6.945	5.275	4.833	3.789	3.170	2.454
12 h	8.517	6.843	5.678	4.791	3.615	2.576	2.526
16 h	8.342	6.732	5.161	4.894	3.849	2.755	2.498
20 h	8.401	6.648	5.501	4.797	3.380	3.007	2.505
24 h	8.443	6.713	5.329	4.901	3.667	2.571	2.536

**Table 6 materials-15-08277-t006:** V precipitation(mg/L).

Time	<0.075	0.075~0.6	0.6~2.36	2.36~4.75	4.75~9.5	9.5~19	19~26.5
4 h	5.562	4.715	4.210	3.732	3.489	2.241	1.985
8 h	5.746	5.213	4.874	4.125	3.889	2.515	2.221
12 h	6.245	5.632	5.187	4.402	3.914	2.626	2.314
16 h	6.454	5.621	5.234	4.698	3.951	2.648	2.369
20 h	6.354	5.598	5.146	4.526	3.879	2.545	2.254
24 h	6.21	5.634	5.098	4.395	3.881	2.556	2.246

**Table 7 materials-15-08277-t007:** Costs of material and transportation.

Item		Unit/(¥/t) *	Cement-Stabilized Limestone		Cement-Stabilized Limestone + 50% Steel Slag		Cement-Stabilized Limestone + 50% Steel Slag + 3% Silica Fume		Cement-Stabilized Limestone + 30% Steel Slag		Cement-Stabilized Limestone + 30% Steel slag + 3% Silica Fume	
			Amount/t	Cost/¥	Amount/t	Cost/¥	Amount/t	Cost/¥	Amount/t		Cost/¥	
Materials	Cement	500	595	297,600	644	321792	644	321,792	609	304,512	609	304,512
	Limestone	100	14,880	1,488,000	7388	738,808	7388	738,808	10,118	1,011,828	10,118	1,011,828
	Steel slag	10	-	-	8702	87,015	8702	87,015	5107	51,073	5107	51,073
	Water	4	634	2221	899	3145	899	3145	682	2389	682	2389
	Silica fume	700	-	-	-	-	261	182,732	-	-	153	107,254
Transportation	tkm	0.35		212,486		412,491		421,628		324,675		330,038
Total		-		2,000,307		1,563,252		1,755,120		1,694,477		1,807,093

Note *: The unit price is from Fangchenggang Construction Bureau in March 2022, China.

## Data Availability

Not applicable.

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
