# Peer review of "Evaluating Cement Treated Aggregate Base Containing Steel Slag: Mechanical Properties, Volume Stability and Environmental Impacts"

_materials, 2022, doi:10.3390/ma15238277_

Round 1
Reviewer 1 Report (Previous Reviewer 1)
Dear Authors
I have no further comments
Best regards
Author Response
Please see the attachment.

Reviewer 2 Report (Previous Reviewer 2)
Dear authors
Some notes made in the initial version of the paper have not yet been included in the resubmission.
Example "The abstract needs to be rewritten, making the information clearer to the reader. The title needs to be revised to make it less speculative".
The item "3.4. Economic and carbon footprint simulation" lacks information on material emissions. The authors only present the energy footprint. Nevertheless, the topic and discussion are about carbon and energy.
Therefore, input values referring to emissions must be explained to the reader.
In general, the authors made a slight improvement in the paper.
The theoretical framework is weak.
There is no correlation between the results with the literature.
The conclusions must present the main results associated with the paper theme "Engineering Properties and Environmental Impacts". I suggest including the numerical values.
Author Response
Please see the attachment.

Reviewer 3 Report (New Reviewer)
The manuscript describes laboratory research on pavement materials destinate to be used in semi-rigid base courses. The pavement materials are cement stabilized aggregates with different incorporations of steel slag aggregates. Engineering properties and environmental impacts were evaluated.
The paper presents an important research topic to contribute to pavement sustainability construction and maintenance.
The paper is recommended to clarify some aspects of the methodology, results and discussion.
1 - The terminology of engineering properties in the title seems very vague.
2 - The mixture gradation in Figure 4 needs more details in the text to be completed. Is the composite gradation the same for all the compositions? (steel slag contents: 0, 30, 50 and 75%). The grain size distributions of the steel slag and limestone aggregates are missing. It isn't easy to understand. In the legend: Composit or Composite?
3 - The description of the test method used for specimen compaction is missing. Impact test? Vibration test?
4 - There is a significant disequilibrium in the paper description: parts of the methodology, results and discussion concerning the evaluation of the environmental impacts are very shorts, and it is impossible to check and follow adequately the methodology and the results presented in Figures 14, 15, 16, and 17.
Author Response
Please see the attachment.

Reviewer 4 Report (New Reviewer)
From a substantive point of view, the article is well written, although there are some minor editorial shortcomings. My comments will therefore be limited to pointing out the corrections that need to be made to the article for it to be publishable.
1. Formula (3) should appear in the text just after it is mentioned in the text, rather than at the very end of the paragraph.
2. In Figure 11a, please change the order of the items in the legend to correspond to either the increasing proportion of steel slag or the order of the curves in the graph.
3. In Figure 13, please add information on the cement content.
Once the above corrections have been incorporated, the article can be printed.
Round 2
Reviewer 3 Report (New Reviewer)
The authors have provided adequate responses to reviewer suggestions. A general improvement in the papers' quality was achieved.
This manuscript is a resubmission of an earlier submission. The following is a list of the peer review reports and author responses from that submission.
Round 1
Reviewer 1 Report
Dear Authors
Your study entitled: “Mechanical and Environmental Evaluation of Applying Steel Slag in Cement Stabilized Semi-rigid Base Course” is rather simple but it has both: scientific and educational value.
I do not have any major concerns about the value and presentation of undertaken experiments. I was pretty surprised that you do not refer to your previous studies as if it was your first time with the recycled concrete issue.
The list of my general comments and some minor editorial concerns are given below.
1. Introduction
Please adjust the way of citing the referenced sources to MDPI templates and try to avoid "cluster citations”. Every referenced paper deserves to be properly introduced to the Reader. Personally, I'd add a bunch of general comments on reuse of metallurgical slag, maybe starting from its direct application in geotechnical engineering to form embankments (which is the least absorbing way of using it, concerning lack of binder cost). Metallurgical wastes (foundry sand, slags) are widely reused in “mining and smelting companies”. Lots of valuable contributions were recently written on this subject, You may check e.g.: https://doi.org/10.3390/ma15072365 and https://doi.org/10.1088/1757-899X/869/3/032004 and other references easy to find in Scopus. It is not mandatory, but it could strengthen your motivation and attract wider attention to your study.
I do not understand why so many of your interest is focused on asphalt mixtures that are not a subject of the current study.
2. Materials and Methodology
This section is cautiously written, concerning material characteristics.
I'd expect a more precise description of strength testing methodology. Please provide information about speed of loading (if possible). An exemplary strain-stress graph would be appreciated, with methodology of its analysis concerning elastic modulus and strength of material.
3. Results and discussions
I observe that you briefly follow the IMRaD structure of scientific contribution. But your study is missing a real Discussion of your results in the light of other Researchers' findings (introduced and referenced in the introductory part). That decreases the clarity of your presentation and a value of your findings, as they are not properly confronted with current "State of the Art" in the discipline.
I have some concerns about the final time of your testing. I understand that 28 days is a standard time in most of codes describing concrete testing worldwide. From my experience, non-standard cement/concrete composites achieve their final capacity and stiffness after 2-3 months. I understand that nowadays, you will not be able to add this information, but you might at least make some reservations concerning the "time issue" and maybe show some prospects for further research.
4. Conclusions
I observe again that your conclusions are proper but a little bit trivial and simply confirming common sense and basic engineering judgement. That is not an objection but just an incentive to continue more profound studies with regard to future practical outcomes.
5. Reference list
The choice of the references is always Authors' right and responsibility. I was pretty surprised that I could not find any self-citations as if Authors never performed similar studies before, But this is not a problem.
I noticed that references to Chinese authors form the majority of the reference list. Most of the references seem to be pretty relevant (maybe except for those related to asphalt mixtures) but I'd strongly recommend to make a short survey in Scopus, looking for more international and recent references. It always widens the group of potential Readers and, last but not least, raises the citing potential of your study.
I marked "major revision" just because of the number of my comments.
Best regards
Reviewer 2 Report
The authors present a study on using granulated slag as a fine aggregate in an established cement base (floor). They make substitutions of up to 75% of natural aggregate.
The article is confusing, with problems with structuring, presentation, and conceptualization.
Based on the method described, it is not possible to replicate the study.
Therefore, the abstract needs to be rewritten, making the information clearer to the reader. The title needs to be revised to make it less speculative.
Explain what macadam cement is.
The authors make generic references in the title, abstract, and objective to investigate physical, mechanical, and environmental impacts. It is known that mechanical properties have several, the same for physical properties. Regarding environmental impacts, which ones will be evaluated? All? just a carbon footprint? Embodied energy? Water or waste footprint? Global warming?
In the abstract, the authors put generic results such as "Basic physical properties manifest that steel slag has the potential of replacing natural aggregate in concrete" or "Balanced performance is obtained when the replacement is 50%" - what is "Balanced performance"? What is this reference?
In the sequence, the authors present a result of treatment with silica fume, so far not mentioned in the abstract "By treated 15 with CH3COOH or adding silica fume, volume expansion of steel slag can be effectively controlled".
Furthermore, "Steel slag has sound economic and potential environmental benefits" is speculative.
The introduction needs to be improved, giving it more density.
Review contradictory information lines 39-40 and lines 46 and 47. Is it 85-90 or 50%?
The authors say that the performance of the samples will be evaluated "through mechanical, volumetric and environmental tests". Detail what the tests would be.
The item Materials and Methods need to be rewritten - making all the processes and procedures of the study clear to the reader.
Translation error "Chemical ingredients of steel slag" Chemical composition...
How important is table 2?
Only in line 106 does the reader discover that the granulated slag is a fine aggregate.
What is the relevance of Figure 5?
Results are also confusing and with little interaction with the literature. In the method, it was not described how the economic analysis and GHG were determined.
The conclusions need to be readjusted.
In the attached text, there are some markings.
Based on the above expose, this reviewer suggests not publishing this paper.

Round 2
Reviewer 1 Report
Dear Authors
I appreciate the time that you devoted to improve your manuscript. Most of my concerns on your study were successfully resolved. I accept your decision to merge results and discussion. I'm looking forward to see your studies on time dependent factors. Please remember that a UCS test, with just a little effort to maintain time, force and shortening of the sample, gives a possibility to establish a modulus that might be later a crucial data for numerical studies.
There are still some minor editorial issues that may be corrected in final proofreading.
1. Please check cautiously the reference format according to MDPI templates. I noticed that your reference [5] should be as follows:
Kongar-Syuryun, C.; Tyulyaeva, Y.; Khairutdinov, A.M.; Kowalik, T. Industrial waste in concrete mixtures for construction of underground structures and minerals extraction. IOP Conf. Ser.: Mat. Sci. Eng. 2020, 869(3), 032004.
2. Please provide sources when introducing every Figure and Table. For example. Content of Table 1 seems to be copied from previous studies without addressing a reference. Figure 1 is probably own photo, but Figure 2 is taken from microscope - please confirm that it is yours or provide reference.
Best regards
Reviewer 2 Report
The authors included some of the suggested suggestions, but these were not enough to improve the quality of the paper.
The main problem was not solved, which is the correct presentation of the Environental Evaluation indicators and the methodology used.
In the item "5. Economic and carbon footprint calculation," the environmental theme appears more appropriately (ie, only the carbon footprint is evaluated).
Authors need to review the title and objectives. The authors also do not describe how the carbon cost indicators were calculated in the methodology.
Therefore, the study is not replicable and should not be published.
